# Coronary Artery Microcalcification: Imaging and Clinical Implications

**DOI:** 10.3390/diagnostics9040125

**Published:** 2019-09-23

**Authors:** Federico Vancheri, Giovanni Longo, Sergio Vancheri, John S. H. Danial, Michael Y. Henein

**Affiliations:** 1Internal Medicine, S.Elia Hospital, 93100 Caltanissetta, Italy; 2Cardiovascular and Interventional Department, S.Elia Hospital, 93100 Caltanissetta, Italy; giova.longo@gmail.com; 3Radiology Department, I.R.C.C.S. Policlinico San Matteo, 27100 Pavia, Italy; sergiovancheri@gmail.com; 4Department of Chemistry, University of Cambridge, Cambridge CB2 1EW, UK; 5Institute of Public Health and Clinical Medicine, Umea University, 901 87 Umea, Sweden; michael.henein@umu.se; 6Institute of Environment & Health and Societies, Brunel University, Middlesex SW17 0RE, UK; 7Molecular and Clinical Sciences Research Institute, St George’s University, London UB8 3PH, UK

**Keywords:** atherosclerosis, coronary microcalcification, atherosclerosis imaging, confocal microcalcification imaging

## Abstract

Strategies to prevent acute coronary and cerebrovascular events are based on accurate identification of patients at increased cardiovascular (CV) risk who may benefit from intensive preventive measures. The majority of acute CV events are precipitated by the rupture of the thin cap overlying the necrotic core of an atherosclerotic plaque. Hence, identification of vulnerable coronary lesions is essential for CV prevention. Atherosclerosis is a highly dynamic process involving cell migration, apoptosis, inflammation, osteogenesis, and intimal calcification, progressing from early lesions to advanced plaques. Coronary artery calcification (CAC) is a marker of coronary atherosclerosis, correlates with clinically significant coronary artery disease (CAD), predicts future CV events and improves the risk prediction of conventional risk factors. The relative importance of coronary calcification, whether it has a protective effect as a stabilizing force of high-risk atherosclerotic plaque has been debated until recently. The extent of calcium in coronary arteries has different clinical implications. Extensive plaque calcification is often a feature of advanced and stable atherosclerosis, which only rarely results in rupture. These macroscopic vascular calcifications can be detected by computed tomography (CT). The resulting CAC scoring, although a good marker of overall coronary plaque burden, is not useful to identify vulnerable lesions prone to rupture. Unlike macrocalcifications, spotty microcalcifications assessed by intravascular ultrasound or optical coherence tomography strongly correlate with plaque instability. However, they are below the resolution of CT due to limited spatial resolution. Microcalcifications develop in the earliest stages of coronary intimal calcification and directly contribute to plaque rupture producing local mechanical stress on the plaque surface. They result from a healing response to intense local macrophage inflammatory activity. Most of them show a progressive calcification transforming the early stage high-risk microcalcification into the stable end-stage macroscopic calcification. In recent years, new developments in noninvasive cardiovascular imaging technology have shifted the study of vulnerable plaques from morphology to the assessment of disease activity of the atherosclerotic lesions. Increased disease activity, detected by positron emission tomography (PET) and magnetic resonance (MR), has been shown to be associated with more microcalcification, larger necrotic core and greater rates of events. In this context, the paradox of increased coronary artery calcification observed in statin trials, despite reduced CV events, can be explained by the reduction of coronary inflammation induced by statin which results in more stable macrocalcification.

## 1. Introduction

Strategies for preventing acute coronary and cerebrovascular (CV) events are based on accurate identification of patients at increased (CV) risk who may benefit from intensive preventive measures [1]. Although the risk stratification using available CV risk scores is useful at a population level, the accuracy of predicting acute events in the individual patient is limited. Current diagnostic strategies are mostly based on the evaluation of symptomatic and asymptomatic patients to detect coronary artery luminal narrowing and myocardial ischemia [2,3,4,5]. However, the importance of the extent of coronary stenosis underlying acute coronary syndrome is debated [6,7,8]. In most instances, an acute coronary event is not due to occlusion at the site of severe stenosis seen on conventional angiography. Rather, the degree of luminal obstruction is a poor predictor of subsequent acute events and most vulnerable plaques are commonly associated with only mild to moderate stenosis [9,10,11,12]. In contrast, thrombosis plays a critical role in the pathogenesis of an acute coronary syndrome [13,14]. The majority of acute coronary events are caused by rupture of the thin cap overlying necrotic core of an atherosclerotic plaque or by plaque erosion, with superimposed thrombus formation [15,16,17]. However, thrombus formation at the site of severe stenosis is more likely to greatly reduce the blood flow, leading to clinical events. Hence, identification of the plaques thought to cause coronary thrombosis, referred to as vulnerable plaques, is essential for optimum acute event prevention. The important role of imaging techniques depends on their ability to identify the morphological and functional characteristics of the vulnerable plaque.

## 2. Pathology

Histological studies have demonstrated that some adverse plaque characteristics, such as a thin fibrous cap, macrophage infiltration, a large necrotic core, microcalcification, neovascularization, intraplaque hemorrhage, and outward arterial remodeling, are consistently associated with plaque rupture and myocardial infarction [18,19,20,21]. Atherosclerosis and calcification are closely related and have traditionally been considered passive, degenerative, the end-stage process of aging. However, several recent studies have demonstrated that they are active, highly dynamic and tightly regulated processes involving cell migration, apoptosis, inflammation, osteogenesis, and intimal calcification, progressing from early lesions to advanced plaques [18,19,22,23,24,25]. Inflammation plays a central role in atherogenesis. In fact, it is involved in all atherosclerosis steps, from plaque formation leading to calcification and rupture [26,27,28]. In most cases, atherogenesis is initiated by endothelial dysfunction which allows subendothelial retention of lipoproteins in arterial regions where the laminar flow is disturbed by arterial branches [19,20]. Inflammatory and immune cells such as macrophages, T cells, and mast cells are then recruited and produce pro-inflammatory mediators and enzymes [29,30]. Macrophages usually have an essential role in the genesis and progression of atherosclerosis [15]. They oxidize and catabolize lipoproteins within the arterial wall. Depending on their amount, oxidized lipoproteins can cause the death of macrophages which finally coalesce into a necrotic core. Vascular smooth muscle cells migrate into the intima and promote the formation of a fibrous cap which is mostly collagen [31]. This lipid-rich necrotic core encapsulated by fibrous tissue constitutes the fibroatheroma, which is generally a stable lesion [18]. Macrophages exert a catabolic effect on the fibrous components of the plaque through the release of metalloproteinases, resulting into plaque cap thinning [28]. Extensive inflammatory process and macrophages infiltration produce the thin-cap fibroatheroma (TCFA), characterized by a large necrotic core separated from the coronary lumen by a thin membrane cap, less than 50 or 60 µm (0.05 or 0.06 mm) thick, which makes the plaque unstable [32,33,34,35,36,37].

The inflammation makes the fibroatheroma hypoxic, resulting in the development of intraplaque neovascularization originating from the vasa vasorum in the adventitia, thus contributing to the destabilization of the plaque [38]. Large lipid tissue and macrophage count are also associated with outward enlargement, or positive remodeling, of the arterial wall [39]. Although the arterial enlargement may be beneficial in avoiding or limiting the luminal stenosis, the plaque is more prone to rupture [21,40,41].

Calcifications in atherosclerotic plaques develop as a healing response to intense local macrophage inflammatory and immune activity, leading also to the osteogenic transformation of vascular smooth muscle cells [42,43,44,45,46]. In lesions with extensive inflammation, macrophage-derived vesicles are released within the necrotic core of the plaque and serve as nucleating sites for calcification [47,48,49]. On the other hand, calcifications themselves stimulate macrophage infiltration [50,51]. As long as inflammation persists there will be subsequent cycles of macrophage infiltration and repair through calcification. The causal relationship between inflammation and calcification has been clinically confirmed by the observation that vascular inflammation, assessed by positron emission tomography, precedes subsequent deposition of arterial calcium within the same vascular site [52]. If the inflammation is reduced, the plaques stabilize leading to macrocalcification [53,54,55].

The relative importance of coronary calcification, whether it exerts a protective effect as a stabilizing force of high-risk atherosclerotic plaque has been debated until recently [54,56]. The extent of calcium in coronary arteries has different clinical implications [45]. Extensive plaque calcification reflects advanced, less inflamed and stable disease [12]. Macrocalcifications make the plaque stable acting as a barrier to limit the spread of inflammation and only rarely result in a rupture. This is confirmed by the observation that the use of statins, which are known to reduce vascular inflammation [57,58], is associated with increased plaque calcification and fibrous cap thickness, resulting in stabilization of the atherosclerotic lesions [59,60,61,62,63,64,65]. Moreover, recent data have shown that the degree of coronary artery calcification is significantly higher in symptomatic patients who had chronic coronary artery disease compared to patients who sustained acute coronary events [43,66,67]. In asymptomatic patients with type 2 diabetes, the plaques more likely to become culprit plaques for acute coronary events over a 8-year follow-up were characterized by larger volume, greater lipid content, and only mild calcification [68]. These observations support the concept that at the level of individual plaque mild calcification predicts subsequent acute events, whereas more advanced calcification has a protective effect.

Unlike largely calcified plaques, microcalcifications, below 60 µm diameter (0.06 mm) [69,70], embedded in the fibrous cap of the fibroatheroma, are associated with increased inflammation and are more frequently observed in patients with acute coronary events [50,66,71]. In combination with other features of plaque vulnerability, such as TCFA, large necrotic lipid core, and extensive presence of macrophages, microcalcifications strongly contribute to plaque instability [45,72,73]. Microcalcifications develop from the aggregation and fusion of individual calcifying extracellular vesicles in areas of inflammation with a large necrotic core [74]. It is thought they may predispose to plaque rupture either through tissue mechanical stress within the fibrous cap of the fibroatheroma [35,75,76,77], and further stimulate inflammation around the plaque [78]. The microcalcific deposits give rise to spotty calcifications, defined as small calcium deposits in the range of 1 to 3 mm involving an arc of about one-fourth of the coronary circumference, embedded in a plaque [49,50,79]. Some of them show a progressive calcification, transforming the early-stage high-risk lesions into stable end-stage macroscopic calcification [80]. Spotty calcifications can also derive from asymptomatic rupture of unstable plaques subsequently healed and have been associated with extensive and progressive coronary atherosclerosis [81,82]. These observations indicate that plaque vulnerability is inversely related to the extent of calcifications.

## 3. Imaging Atherosclerosis

In addition to CV risk assessment using risk factors, imaging is used to directly detect the presence and extent of atherosclerotic disease. Microcalcification and inflammation play a key role in plaque rupture, therefore representing important potential imaging targets. Although the greatest efforts have been undertaken to identify and treat the TCFA with signs of inflammation as high-risk lesions for acute coronary events, it is now well established that most vulnerable plaques do not cause cardiac events [7,83,84]. The majority of unstable plaques undergo progressive transformation from high-risk lesions with microcalcifications to more stable macroscopic calcification. Others likely heal following asymptomatic rupture and stabilize with time [85]. According to these observations, imaging aimed at the detection and treatment of individual plaques has a limited impact on prevention. However, unstable plaques do not occur in isolation. Some studies suggest that plaque destabilization occurs at multiple distant sites throughout the coronary and carotid vascular beds [86,87,88]. Hence, their detection may serve to identify high-risk individuals with more extensive atherosclerotic disease activity, who may benefit from systemic therapy [89]. Atherosclerosis imaging is accordingly evolving from anatomical to metabolic imaging, thus providing insight into the underlying vascular inflammation of patient.

## 4. Imaging Plaque Morphology

Direct visualization of coronary atherosclerotic lesions is currently performed using angiography, computed tomography (CT) coronary artery calcium (CAC) scoring, coronary computed tomography angiography (CCTA), intravascular ultrasound (IVUS), and optical coherence tomography (OCT), each providing different information about plaque morphology.

A coronary angiogram is routinely used to assess the degree and extension of arterial stenosis. Although it can detect extensive superficial calcium plaques, its sensitivity for smaller lesions is less than 50%, being to some extent operator-dependent [90]. Moreover, the relationship between the severity of stenosis and vulnerable plaques is uncertain. When the presence of vulnerable lesions is assessed by more specific intracoronary imaging, such as IVUS and OCT, a greater number of TCFA is detected in non-severe than in severe stenotic arterial lesions [11]. However, the underlying lesion morphology at the site of severe stenosis is more vulnerable and thrombus formation following the rupture more likely to further limit the blood flow leading to clinical events [91].

Non-contrast cardiac-gated computed tomography (CT) is extensively used to calculate the CAC score which correlates with the total coronary atherosclerotic burden and has been found as a strong predictor of patient outcomes [92,93,94,95,96]. Guidelines recommend CAC assessment using the Agatston scoring system to improve CV risk prediction in asymptomatic individuals at intermediate risk as well as in diabetics [1,3,4,5]. Recent studies have shown that CAC volume and density have different relationships with patients’ outcomes. While CAC volume is a direct predictor of CAD, density is inversely associated with acute coronary events [97,98]. These observations may suggest that a higher calculated volume of calcium is associated with extensive coronary atherosclerosis, whereas higher calcium density indicates more stable plaques. However, although the extent of coronary calcification can be accurately quantified, the current spatial resolution of 64-slice CT imaging cannot identify microcalcifications or distinguish actively inflamed from stable atherosclerotic calcifications.

In patients with stable coronary artery disease at low-intermediate risk, the addition of intravenous contrast agent to coronary CT provides non-invasive, highly accurate insight into the extent of coronary calcification and detection of obstructive coronary stenosis and high-risk coronary plaque morphology [99,100] (Figure 1 and Figure 2). However, accurate quantification of various plaque components provided by CCTA is limited [101]. Microcalcifications and TCFA cannot be detected because their dimension is about ten times below the CCTA scan spatial resolution (about 0.5 mm) [102]. Hence, the anatomic assessment of the coronary plaques provided by CCTA, although a good predictor of the global risk of acute events is not useful for identifying vulnerable lesions prone to rupture in a specific lesion [103,104,105]. Overall, there is no evidence that CCTA plaque assessment improves the prediction of acute coronary event risk compared to established risk factors [106].

Intravascular imaging such as IVUS or OCT enables the clinician to gain insight into the arterial wall for the detection and quantification of calcium within a plaque and to use this information for patient care [107,108] (Figure 3). IVUS has high sensitivity and specificity for detecting large dense calcific plaques or spotty calcifications [80,81]. However, its axial resolution, in the range of 150–200 µm (0.15–0.20 mm), is not sufficient to visualize microcalcifications or the thin-cap fibroatheroma which are usually smaller than 60 µm (0.06 mm). Since IVUS cannot penetrate calcium, its assessment is limited to arc and length. Another limitation is the inability to assess the composition and inflammation state of the fibrous cap [109].

OCT is a light-based imaging modality, analogous to ultrasound, which measures the time delay of optical echoes reflected by the arterial structures, providing high resolution, cross-sectional images of their structure [37,110,111]. Compared to IVUS, OCT has about ten times higher resolution, between 10 and 20 µm (0.01–0.02 mm). Unlike IVUS, OCT can penetrate calcium and assess its thickness, area, and volume, thus having the potential to characterize the details of coronary calcification (Figure 4). According to the OCT imaging of calcium arc, defined as the widest angle in which the calcifications are detectable, the coronary calcifications are classified as macrocalcifications with a calcium arc >90°, spotty calcification between 90° and 22.5°, and microcalcifications with a calcium arc <22.5° [112]. OCT is capable of quantifying the presence of macrophages in the atherosclerotic plaque with a high degree of positive correlation with histology, demonstrating high sensitivity (>85%) and specificity (89%) [113,114,115]. This provides direct evidence of the level of plaque inflammation. Recent OCT studies in patients with stable CAD have demonstrated that the contemporary presence of macrophages and microcalcifications in the same plaque (co-localization, when the reciprocal distance is smaller than 1 mm) is associated with a more vulnerable plaque and with systemic features of atherosclerosis such as increased carotid intima-media thickness [73,78]. The same patients showed less advanced coronary artery stenosis, thus suggesting that the co-localization of macrophages and microcalcifications indicates an early phase of the atherosclerotic process which may progress into further calcification and inflammation. Taken together these observations indicate that OCT can provide both morphological and assessment of the level of disease activity, thus identifying patients at higher risk for subsequent coronary events [116].

## 5. Imaging Disease Activity

Cardiac magnetic resonance (CMR) and positron emission tomography (PET) provide an assessment of microcalcification and inflammation.

Due to limited spatial resolution (1.3–1.8 mm) and long scan duration, CMR has limited indications in clinical practice, whereas it has important research interest for future applications. Preliminary studies have shown that CMR can identify some unstable coronary plaque characteristics [105]. Hyperintensity plaques on non-contrast T1-weighted imaging were identified as high-risk plaques, validated by intravascular imaging [117,118,119,120]. Contrast-enhanced CMR plaque imaging provides a higher spatial resolution. The accumulation of the gadolinium-based contrast agent has been associated with macrophage accumulation and intraplaque hemorrhage, as confirmed by OCT [121,122,123,124]. The absence of ionizing radiation or potentially nephrotoxic contrast agents, could make CMR a non-invasive imaging modality to monitor the atherosclerotic disease activity.

PET provides a non-invasive imaging method for the assessment of the underlying biological activity within the arterial wall [105,125]. Specific radioligands are used, targeting microcalcification (^18^F-sodium fluoride) and macrophages (^18^F-fluorodeoxyglucose and somatostatin receptor ligand). To provide anatomic details, PET scans are currently combined with CT or MR. Recently developed hybrid scanners PET/MR offer the possibility to simultaneous assessment of disease activity and morphological information with a lower radiation exposure compared to PET/CT [124,126].

^18^F-sodium fluoride (^18^F-NaF) was originally studied to identify bone metastasis. Fluoride ions are incorporated into hydroxyapatite, which is a central component of osteogenic mineralization [127]. Coronary atherosclerosis is directly connected to macrophages osteogenic activity in the early stages of atherosclerosis, which results in microcalcifications [42]. This allows ^18^F-NaF to detect active microcalcification area beyond the resolution of CT scan [128,129,130,131]. The stronger affinity of the radioligand with newly formed hydroxyapatite compared to the old crystals makes it possible to distinguish between actively inflamed coronary calcifications from stable ones. This is confirmed by the observation that large areas of coronary calcium detected by CT scan do not show increased ^18^F-NaF uptake. Conversely, regions with absent or minimal CT calcium demonstrate intense ^18^F-NaF uptake [132,133]. This may explain the lack of correlation between ^18^F-NaF atherosclerotic plaque uptake and CAC score observed in high-risk individuals [134].

^18^F-fluorodeoxyglucose (^18^F-FDG) is a glucose analogue which accumulates mostly in macrophages due to their high demand for glucose. It is a marker of vascular inflammation and macrophage burden and is extensively used for malignancy staging. Several studies conducted in carotid arteries have demonstrated the systemic nature of atherosclerosis [127,135,136]. Its use in the assessment of coronary plaque inflammation is limited by the close proximity to myocardial tissue, which has a high affinity to tracer uptake due to its high glucose metabolism. This obscures the coronary visualization, thus limiting accurate plaque analysis.

## 6. Confocal Imaging of Microcalcifications

Besides the above-mentioned histological visualization methods, fluorescence-based techniques can provide imaging of micro-calcifications with high sensitivity, selectivity, and resolution. Confocal microscopy, routinely applied to biological imaging, can penetrate deep inside heart tissues to provide clearer images of intricate structures [137]. Unlike other assessment methods where the spatial resolution cannot be decreased beyond 10 µm, the resolution in confocal microscopy can reach ~500 nm. In confocal microscopy, a tightly focused laser spot is scanned across a fluorescently labeled sample and the collected emission is filtered through a micro-meter-sized pinhole to exclusively allow the in-focus fluorescence reach a sensitive detector and reject all out-of-focus fluorescence contributed from other imaging planes. As such, confocal microscopy allows fine sectioning of samples to provide clear images with high signal-to-noise ratio. Alhough superior to other imaging techniques, confocal microscopy requires the sample to be immuno-stained with an appropriate fluorescent dye with low photo-bleaching kinetics to allow unperturbed imaging over long durations. This might represent a challenge to pathologists using Hematoxylin and Eosin (H&E) staining for their assessments. Furthermore, the field-of-view in confocal microscopy is limited by the magnification of the imaging objective and the linear range of the laser scanners to approximately 250 µm. As means to mitigate this shortcoming, it is possible to identify Regions of Interest (ROIs) using the low-resolution methods outlined above and, subsequently, use confocal microscopy to acquire high-resolution imaging of these ROIs.

## 7. Conclusions

Recent developments in the understanding of the pathophysiology of plaque vulnerability have demonstrated the central role of inflammation and microcalcification. The need to identify early stages of unstable lesions has shifted the patients’ risk factor assessment for CV events to direct imaging-based detection of plaque morphology and disease activity. Further means of accurate study and assessment of vascular microcalcification is expected to have more profound clinical impact on disease prevention.

## Figures and Tables

**Figure 1 diagnostics-09-00125-f001:**
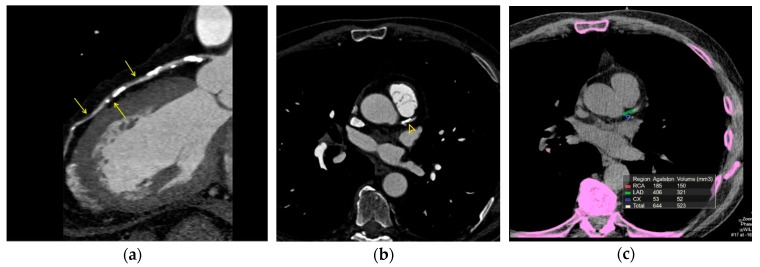
(**a**). Coronary computed tomography angiography. 79-year-old hypertensive patient demonstrating extensive calcified plaques of left anterior descending (LAD) coronary artery. Lipid-rich (low CT attenuation) plaques are also shown (arrows). (**b**). Same patient. Axial view. The calcific plaque involves the origin of the LAD coronary artery (yellow arrowhead). (**c**). Same patient. Agatston CAC score >400, indicating extensive calcification, high CV risk and high likelihood of significant coronary stenosis.

**Figure 2 diagnostics-09-00125-f002:**
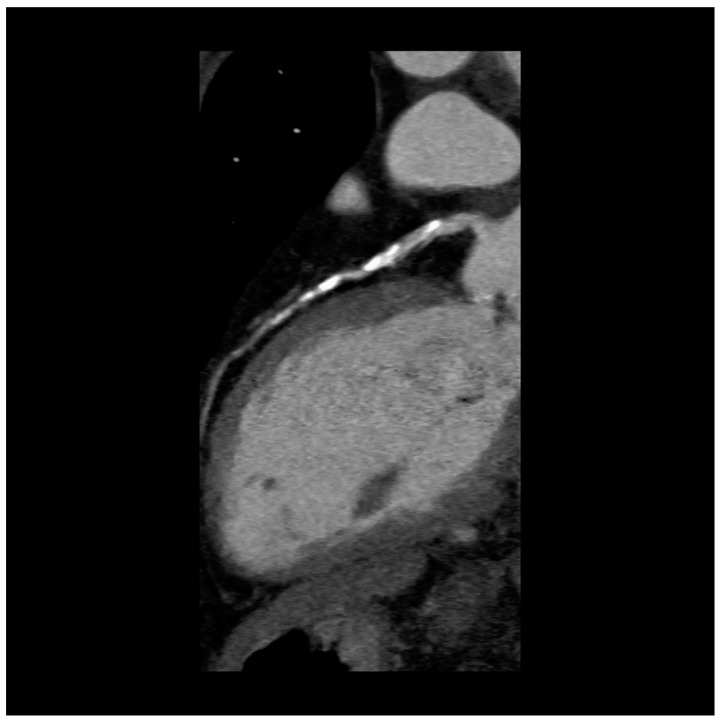
Coronary computed tomography angiography. 59-year-old woman with type-2 diabetes, symptomatic for stable angina, showing extensive calcified plaques of the LAD coronary artery.

**Figure 3 diagnostics-09-00125-f003:**
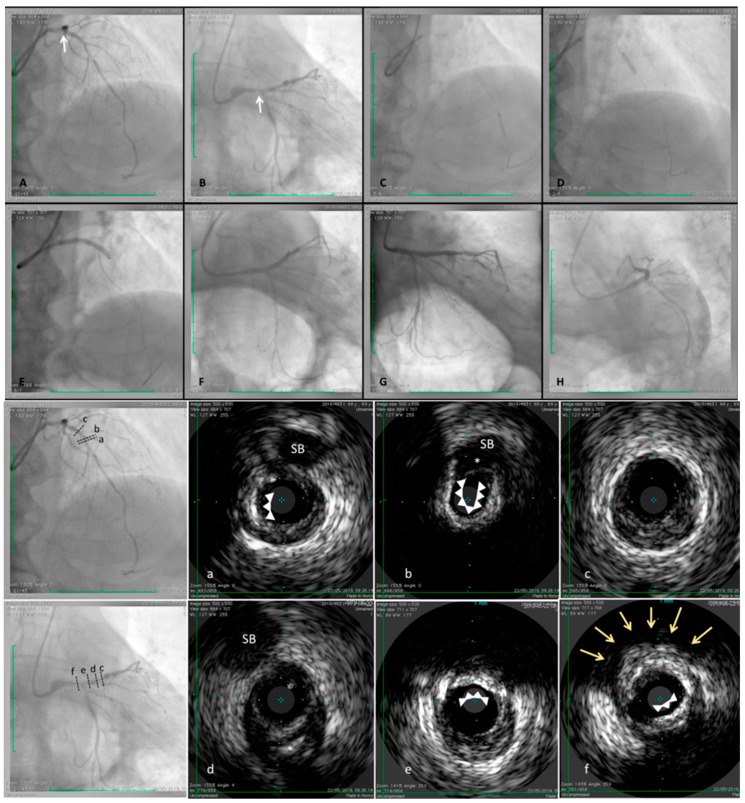
Angiographic assessment by IVUS images in a fibro-calcific plaque involving left anterior descending (LAD) and left main (LM) coronary arteries. 69-year-old male patient, hypertensive, obese, previously treated by percutaneous coronary intervention (PCI) and drug eluting stent (DES) implantation to right coronary artery (RCA) because of inferior STEMI occurred. A staged procedure was scheduled to treat distal and mid left anterior descending (LAD) and to perform an IVUS assessment at proximal LAD and left main LM trunk. After stenting distal and mid segment (**C** and **D**), followed by a properly post-dilation, IVUS pullback was performed. The images showed a severe and calcific stenosis involving diagonal ostium (cross section “**a**” and “**b**”), an almost complete calcific ring (cross section “**a**” and “**b**”, white arrows) is stopped just at side branch take-off (cross section “**b**”, white asterisk). Proximally, a fibrotic prevalent plaque was detectable (cross section “**c**”). More proximally, fibrotic plaque associated by calcific spot was identified (cross section “**d**”, white asterisk). Immediately distal the bifurcation side (LM–LAD–LCx) another calcific deposits, forming an arch (around 110°), was detected (cross section “**e**”) and the site of distal LM truck a superficial calcific deposit (cross section “**e**”, white arrows) and a deep calcium nodule were detected (cross section “**f**”, yellow arrows), even visible by angiography (**A** and **B**, white arrows). After adequate pre-dilation, a long stent was implanted to LM-LAD (**E**) and a satisfactory result was reached (**F**–**H**).

**Figure 4 diagnostics-09-00125-f004:**
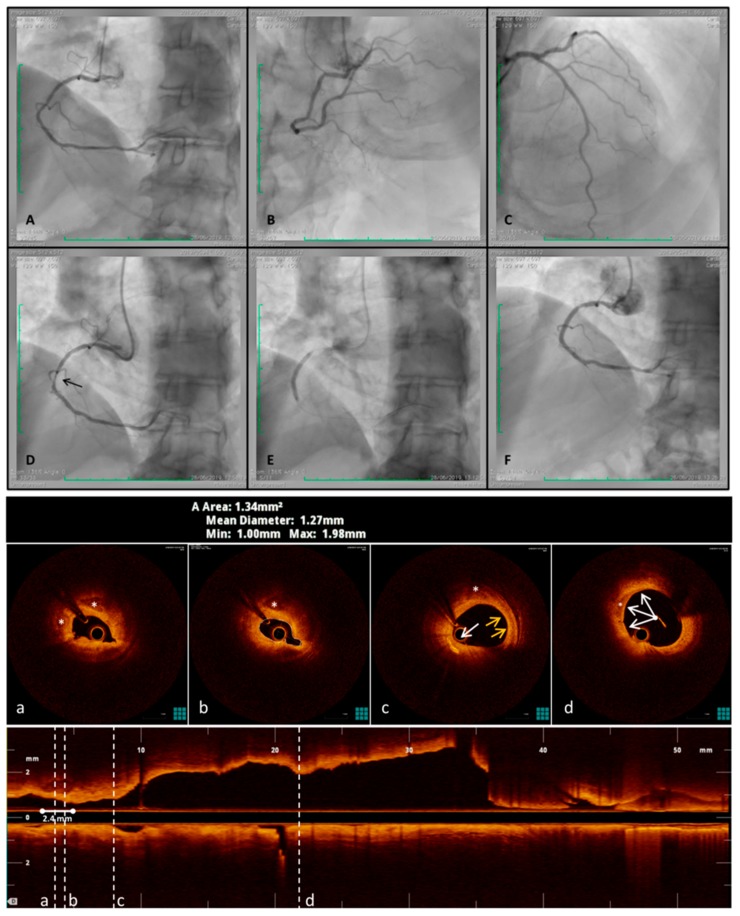
OCT overcomes angiography limits during an acute coronary syndrome with transient ST-elevation in a symptomatic patient. This is a clinical case that can well show plaque activity in an Acute Coronary Syndrome (ACS). A 66-year-old man, smoker, hypertensive, presented chest pain at Emergency Department. The EKG showed a transient inferior ST-elevation associated at high-sensitive Troponin increased. An early (<2 h) coronary angiography was performed but, no significant lesions were detected (**A**–**C**). Optical Coherence Tomography (OCT) was executed and a severe spasm occurred immediately after guide-wire crossed the lesion (**D**, black arrow). Nitroglycerine administration did not reverse the intense spasm and the intravascular imaging show a very tight lesion with a minimum lumen area of 1.34 mmq (cross section “**b**”). In the atheroma two spotty calcifications (cross section “**a**” and “**b**”, white asterisks) and, proximally, activated macrophages (cross section “**c**”, white arrow), TCFA (yellow arrow) and lipid pool (white asterisk), were detected. Moreover, other two smaller and superficial calcifications (cross section “**d**”, white arrow and asterisk), forming, joined an arch (<90°), were identified. A stent was placed and optimized, guided by OCT features, reaching a good angiography result (**E** and **F**).

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
