# Peer review of "Coronary Artery Microcalcification: Imaging and Clinical Implications"

_diagnostics, 2019, doi:10.3390/diagnostics9040125_

Round 1

Reviewer 1 Report

The study of coronary lesions with the correct identification of characteristics at high risk for the development of cardiovascular events is one of the great challenges in cardiovascular imaging. The various techniques available provide us with different information about atherosclerotic plaques and their integration is often needed to determine the lesions to be treated. Moreover, invasive approaches are usually necessary, although recent advances in the field of merging non-invasive strategies able to combine anatomical features and biological activity of plaques represent an important perspective.

The aim of this review is to give us a state of the art of the pathology, the microscopic and functional characteristics and the techniques used to analyze coronary lesions and to identify high risk plaques. The Authors conducted an accurate study of literature and precisely highlighted the pathophysiological mechanism and the high risk aspects of coronary lesions, emphasizing the strengths and weaknesses of every technique available.

The conclusions lead the reader to reflect on the current cardiovascular risk assessment, underlining the need for integration with imaging techniques that give us morphological and functional aspects of atherosclerotic plaques.

I only have some minor points:

in the section "Imaging plaque morphology" the Authors could refer to the new 2019 ESC guidelines on the diagnosis and management of chronic coronary syndromes, in particular when talking about CT indications. Lines 204 to 220: they are part of the caption of figure 3. Lines 245 to 259: they are part of the caption of figure 4. Line 252: verb to show should be conjugated to the past.

Author Response

in the section "Imaging plaque morphology" the Authors could refer to the new 2019 ESC guidelines on the diagnosis and management of chronic coronary syndromes, in particular when talking about CT indications.

R: In the section "Imaging plaque morphology", about CT indications, lines 177, 178, a sentence and the reference to ESC 2019 have been added.

Lines 204 to 220: they are part of the caption of figure 3.

R: All the caption of the figures have been made clearer.

Lines 245 to 259: they are part of the caption of figure 4.

R:All the caption of the figures have been made clearer.

Line 252: verb to show should be conjugated to the past.

R:Line 252, the verb to show coniugated to the past.

Reviewer 2 Report

This is a review on the calcification of the plaque farmed into the coronary artery with its relation to cardiovascular risk and their identification by imaging technique which is well written but too compact.

However, I would like to suggest the authors to elaborate the content with an explanatory narrative which may increase overall discussion by ~15%.

For example at line 109 “Moreover, recent data have shown that the degree of coronary artery calcification is significantly higher in symptomatic patients who had chronic coronary artery disease compared to patients who sustained acute coronary events” – authors may choose to elaborate the implications of such a phenomenon.

I also have a few minor concerns about the followings.

Line 122. “calcifications defined as small calcium deposits in the range of 1 to 3 mm or involving an arc of about” It seems “or” should be removed.

Page 5, 6 and 7 contains 3 different pictures but all are labeled as figure 1.

Page 9. They have presented Figure 3. After that, they have discussed about Figure 1A-H. Most likely they discussed this figure 3. It is also not clear they have a reference for this story, or it is an unpublished result.

Page 11. After presenting Figure 4, they discussed the picture mentioning it as Figure 2A-F. Is it too a published or unpublished work- please specify? They could introduce subheadings for these discussions.

Author Response

This is a review on the calcification of the plaque farmed into the coronary artery with its relation to cardiovascular risk and their identification by imaging technique which is well written but too compact.

However, I would like to suggest the authors to elaborate the content with an explanatory narrative which may increase overall discussion by ~15%.

For example at line 109 “Moreover, recent data have shown that the degree of coronary artery calcification is significantly higher in symptomatic patients who had chronic coronary artery disease compared to patients who sustained acute coronary events” – authors may choose to elaborate the implications of such a phenomenon.

R: An explanatory narrative has been added in the lines 111-116, about the implications of mild vs. extensive plaque calcification

Line 122. “calcifications defined as small calcium deposits in the range of 1 to 3 mm or involving an arc of about” It seems “or” should be removed.

R: Line 126, the "or" removed

Page 5, 6 and 7 contains 3 different pictures but all are labeled as figure 1.

R:The labels of the figures have been made more clear.

Page 9. They have presented Figure 3. After that, they have discussed about Figure 1A-H. Most likely they discussed this figure 3. It is also not clear they have a reference for this story, or it is an unpublished result.

R:The labels of the figures have been made more clear.

Page 11. After presenting Figure 4, they discussed the picture mentioning it as Figure 2A-F. Is it too a published or unpublished work- please specify? They could introduce subheadings for these discussions.

R:The labels of the figures have been made more clear.